# The Endothelial Landscape and Its Role in Von Hippel–Lindau Disease

**DOI:** 10.3390/cells10092313

**Published:** 2021-09-04

**Authors:** Isabel de Rojas-P, Virginia Albiñana, Lyudmyla Taranets, Lucía Recio-Poveda, Angel M. Cuesta, Nikita Popov, Thales Kronenberger, Luisa M. Botella

**Affiliations:** 1Centro de Investigaciones Biológicas Margaritas Salas, 28040 Madrid, Spain; iderojas@usal.es (I.d.R.-P.); vir_albi_di@yahoo.es (V.A.); luciarecio@hotmail.com (L.R.-P.); angcuest@ucm.es (A.M.C.); 2Centro de Investigación del Cáncer, Instituto de Biología Molecular y Celular del Cáncer, 37007 Salamanca, Spain; 3CIBERER, Centro de Investigación Biomédica en Red de Enfermedades Raras, ISCIII, 28029 Madrid, Spain; 4Department of Internal Medicine VIII, University Hospital Tübingen, Otfried-Müller-Strasse 14, 72076 Tübingen, Germany; ltaranec@gmail.com (L.T.); nbpopov@googlemail.com (N.P.); kronenberger7@gmail.com (T.K.); 5Departamento de Bioquímica, Facultad de Farmacia, Universidad Complutense de Madrid, 28040 Madrid, Spain

**Keywords:** VHL, BOECs, endothelial cells, ROS, angiogenesis, wound healing, cell adhesion, VEGF pathway

## Abstract

Von Hippel–Lindau disease (VHL) is a rare hereditary disease characterized by the predisposal to develop different types of highly vascularized tumors. VHL patients carry a *VHL* mutation that causes partial lack of functional VHL protein (pVHL) in all cells, and a total lack thereof in cells harboring a second hit mutation. Absence of pVHL generates a prolonged state of pseudo-hypoxia in the cell due to accumulation of hypoxia inducible factor, an important transcription factor regulating pro-tumorigenic genes. The work here presented focuses on characterizing the endothelium of VHL patients, by means of blood outgrowth endothelial cells (BOECs). Transcriptome analysis of VHL-derived BOECs, further supported by in vitro assays, shows that these cells are at a disadvantage, as evidenced by loss of cell adhesion capacity, angiogenesis defects, and immune response and oxidative metabolic gene downregulation, which induce oxidative stress. These results suggest that the endothelium of VHL patients is functionally compromised and more susceptible to tumor development. These findings contribute to shedding light on the vascular landscape of VHL patients preceding the second hit mutation in the *VHL* gene. This knowledge could be useful in searching for new therapies for these patients and other vascular diseases.

## 1. Introduction

Von Hippel–Lindau disease (VHL) is a rare hereditary disease of autosomal dominant inheritance affecting 1 in every 36,000 births. VHL disease is characterized by the predisposal to develop different types of highly vascularized tumors, hemangioblastomas (HB) of the Central Nervous System (CNS) or retina, malign clear cell renal cell carcinomas (ccRCC) and phaeochromocytomas, and lesions that include cysts of the pancreas and kidneys [1,2]. The molecular genetics of VHL disease have been well characterized and, consequently, different subtypes of the disease are linked to a diverse range of mutations, comprising inactivating deletions, frameshifts, and missense mutations in the *VHL* gene [3]. *VHL* is a tumor suppressor gene encoding for two isoforms of VHL protein (pVHL), both playing an important role in regulating the hypoxia response in all cell types, being considered of great importance in maintaining cell homeostasis [4].

Most VHL patients inherit a germline *VHL* mutation from an affected parent and a wild-type copy from the unaffected parent, resulting in heterozygotes for *VHL* gene in all cell types (first hit mutation), although 20% of cases are caused by a sporadic de novo mutation [4]. VHL disease follows Knudson’s two-hit hypothesis of hereditary tumorigenesis: cell types suffering a somatic inactivation of the wild-type allele (second hit mutation) are prone to tumor formation [5,6]. pVHL forms a complex with other proteins (Elongin B and C, CUL2, Rbx1, and E2) that functions as a ubiquitin-protein ligase (E3), targeting the α subunits of the hypoxia inducible factor (HIF) [7]. During physiological normoxic conditions, hydroxylation of prolyl residues promotes binding of HIF-α subunits to pVHL, which leads to HIF ubiquitination and its subsequent proteasomal degradation [8]. Under hypoxic stress, however, lack of oxygen inhibits activity of the prolyl hydroxylase, in charge of HIF-α hydroxylation (PHD2), impairing HIF-α-pVHL interaction. When this happens, HIF-α may enter the nucleus and heterodimerize with HIF-1β [9]. The HIF heterodimer product has been shown to bind hypoxia-responsive elements (HREs) in over 500 different gene loci, most containing consensus core HIF-binding motif RCGTG [10]. HIF’s binding to promoters and enhancers of target genes regulates the hypoxic survival response [11]. These findings led to the awarding of the 2019 Nobel Prize for Medicine to key contributors in the field [12].

Therefore, VHL patients have either a partial lack of pVHL or lack of functional pVHL in all cells due to their heterozygosis in *VHL*, and a total lack of functional pVHL in cells suffering the second hit mutation. As a consequence of this second mutation, a continuous state of pseudo-hypoxia is generated: although cells are not lacking oxygen, the absence of pVHL leads HIF-1α and/or HIF-2α to accumulate and enter the nucleus, where, upon dimerization with HIF-1β, it promotes the sustained expression of pro-tumorigenic factors such as vascular endothelial growth factor (VEGF), platelet-derived growth factor (PDGF), erythropoietin (EPO), and transforming growth factor alpha (TGF-α). The uncontrolled overexpression of HIF’s target genes in *VHL* (−/−) cells ultimately leads to rapid proliferation and angiogenesis, promoting tumor formation in VHL patients [13,14].

Researchers studying VHL disease have previously highlighted the relevance of the VHL rare disease as a model for a better understanding of cancer biology [15]. Despite the translational efforts of researchers and physicians over the last decades, VHL patients lack a definite systemic therapy, and local recurrent surgeries remain their main line of treatment. The most common surgeries are in the CNS (spinal cord, brain trunk, and cerebellum), followed by surgeries of renal carcinoma [4]. The CNS surgeries decrease the health span of patients due to sequelae, while surgery of renal carcinomas often remain insufficient, turning this tumor into a common cause of death in these patients due to its metastatic nature [16,17].

CNS-HBs are derived from the stem cell hemangioblasts, giving rise to endothelial, mural, and stromal cells. The mixed nature of the tumors makes it difficult to find the proper cell type to target when trying to obtain a genetic model representative of the disease [18]. In addition, the lack of an animal model of the VHL disease is a major drawback in discovering alternative therapies for VHL patients. To date, there is still no suitable model to study HBs of the CNS, and only a conditional *VHL* knockout has been achieved in the mouse retina [19]. CcRCC, another common manifestation of VHL disease, also currently lacks an animal model. Renal and pancreatic cysts, and liver cavernous HBs, are the only VHL symptoms that have been reproduced in mice [18]. The difficulty of obtaining a VHL mouse model relies in the lethality of a homozygote knockout, while the heterozygote *VHL* (+/−) does not present HBs or any of the recurrent tumors from VHL patients.

For these reasons, at this moment, in vitro cellular assays constitute the most reproducible and practical approach in the search for new pharmaceutical treatments to treat and, at least, reduce the number of surgical interventions of VHL patients. In recent years, a new line of research has paved the way for drug repurposing of β-blockers in treating VHL disease, first with the non-specific β-blocker propranolol [20,21] and more recently with a β2-specific antagonist [22]. The work of Albiñana et al. (2020) [23] first proved effectiveness of β2 blockers on ccRCC cell lines. Moreover, it has been used “off label” in a few cases to successfully treat patients. Despite how far we have come in our understanding of VHL syndrome, most efforts have focused on its cancer biology, but little is known about the state of the patients prior to disease onset. The work here presented aims to comprehend the basal physiology of VHL patients preceding tumor development, focusing on the endothelium given the vascular nature of the disease and by means of blood outgrowth endothelial cells (BOECs) as a novel in vitro tool for studying VHL disease.

## 2. Materials and Methods

### 2.1. Cell Culture

For the growth and expansion of BOECs, 50 mL of periphery blood was extracted from volunteers after reading and signing written consent. Blood samples were subjected to a Ficoll density gradient, for which samples were first diluted 1:1 in HBSS and then carefully transferred to 50 mL Falcon tubes containing 25 mL of Lymphoprep^TM^ (PromoCell, Heidelberg, Germany). These were centrifuged at 900× *g* for 20 min with no brake. Peripheral blood mononuclear cells (PBMCs) halos were then collected, washed, and seeded on previously collagen-coated 6-well plates, in Endothelial Growth Medium-2 (EGM-2TM) (Lonza, Walkersville, MD, USA) supplemented with 10% FBS, 2 mM L-glutamine, and 100 U/mL penicillin/streptomycin (GIBCO, Grand Island, NY, USA).

### 2.2. Tubulogenesis In Vitro Assay

24-well-plates were previously coated with 100 μL Matrigel Matrix (BD Biosciences, Bedford, MA, USA) diluted 1:2 in serum-free DMEM plain medium (GIBCO). Then, 1 × 10^5^ BOECs were seeded and cultured in EGM for 6 h. Images were taken at different times using an Olympus digital camera, and quantification of closely connected cells of tubules was performed applying the GIMP 2.10 software (UC Berkeley, Berkeley, CA, USA).

### 2.3. Wound Healing Assay

Cell motility was assessed by wound healing recovery in vitro assay. Upon reaching confluence in a 24-well plate, the monolayer of BOECs was scratched using a T-200 tip, making discontinuity gaps (wounds) within each plate. Then, cell debris was washed out with PBS and fresh EGM was added. Endothelial cells (ECs) migration into the denuded area was monitored at 0- and 6.5-h post-wounding. Images were taken with an Olympus digital camera, and GIMP 2.10 software (UC Berkeley) was used to quantify the distances of the wound healing process.

### 2.4. Flow Cytometry

BOECs were seeded in 24 well-plates until reaching confluence and were then collected for incubation in suspension with mouse monoclonal antibodies against human Endoglin (P4A4), VE-cadherin (Sigma, St Louis, MO, USA), VEGFR2/KDR (R&D, Abingdon, UK), or rabbit monoclonal antibody against human von Willebrand Factor (VWF) (Abcam, Cambridge, UK). For VWF, since it is localized in the cytoplasm, cells were previously permeabilized in methanol for 30 min to allow the antibody to enter the cell. A negative control was included using an irrelevant isotype control antibody XP3/63 (IgG1). After washing the cells, they were incubated with Alexa Fluor 488 goat anti-mouse (A-11001 Molecular Probes, Thermo Fisher Scientific, Waltham, MA, USA) for Endoglin, VE-cadherin and VEGFR2/KDR detection, or Alexa Fluor 488 goat anti-rabbit IgG (11008 Molecular Probes) for VWF detection. Fluorescence intensity was measured with an EPICS XL flow cytometer (Beckman Coulter, Brea, CA, USA). A minimum of 10^4^ cells were counted for each experimental point.

### 2.5. Immunofluorescence Microscopy

Immunofluorescence analyses were performed to characterize the isolated BOECs using specific ECs marker VWF, and to stain for the pVHL (Novus, Oxon, UK). For this purpose, 5 × 10^3^ BOECs were seeded on sterile 13 mm diameter coverslips (VWR international, Radnor, PA, USA) placed at the bottom of a 24 well-plate, previously coated with collagen. On the next day, cells were washed with PBS and fixed with 3% PFA for 10 min at RT. After two PBS washing steps, samples were incubated with blocking solution (1% goat serum and 1% BSA, in PBS) for 1 h at RT. Then, cells were incubated overnight at 4 °C with rabbit anti-VWF antibody (1:50) (Abcam). Following this, cells were washed thoroughly four times with PBS and incubated for 1 h at RT with goat anti-rabbit IgG (H + L)-Alexa fluor 568 conjugate antibody (1:200) (Thermo Fisher Scientific). Finally, cells were washed with PBS and coverslips were mounted on glass slides using Prolong-DAPI mounting media (Molecular Probes), and 40× confocal images were taken using the fluorescence confocal microscope Sp5 (DMI6000 CS Leica Microsystems, Wetzlar, Germany). Green and blue channels represent VWF and DAPI stains, respectively.

### 2.6. RNAseq

Early BOECs passes were grown to confluency, and RNA extraction and library preparation were completed following the NEBNext Ultra RNA Library Prep Kit for Illumina (New England Biolabs, Ipswich, MA, USA). Sample integrity was confirmed by 28S/18S RNA ratios above 1.3 upon RNA gel electrophoresis. After library preparation, Illumina NextSeq 550 system was used for mRNA sequencing.

Sequencing reads were pre-processed by means of a pipeline that used FastQC to assess read quality, and Cutadapt v1.6 to trim sequencing reads, eliminate Illumina adaptor remains, and discard reads <30 base pairs. The human genome GRCh38 and Gencode Release 31 annotations were used to map and count sequence reads. The reads were mapped to the GRCh38 reference genomes using STAR v2.6.2h. Only the reads that uniquely mapped to exonic regions were counted, and reads that mapped to gene overlapping regions were excluded. Raw counts per gene were converted to the log values of counts per million (LogCPM), and genes with no counts in more than two-thirds of the samples within the same experimental group were disregarded for the following steps. Differential expression analysis was performed using the edge R package and an experimental-based design matrix comparing the VHL samples against the control group. Further, differentially expressed genes were identified at FDR of <0.05 and log2 fold change >1.

Integrative analysis was performed manually in R (v3.6) to obtain lists of genes that overlap to publicly available datasets of interest. Briefly, a list of differentially expressed genes up- or down-regulated (DEGs) underwent an unbiased approach to characterize our datasets’ using enrichment analysis. Enrichment of transcription factors and metabolic pathways (exploring KEGG and WikiPathways), and gene ontology terms on our datasets were carried out using WebGesalt server, with standard settings, using both Over Representation and Gene-set enrichment modes with default settings. Information from all significantly enriched terms (adjusted *p*-value as represented by FDR < 0.05) from all datasets was pooled to interpret the evaluated datasets. RNAseq data is available as GEO dataset GSE181417.

### 2.7. Real-Time RT-PCR (qPCR)

Total RNA was extracted from BOECs using Nucleo Spin RNA kit (Macherey-Nagel, Düren, Germany). One microgram of total RNA was reverse transcribed in a final volume of 20 μL with the First Strand cDNA Synthesis Kit (Roche, Mannheim, Germany) using random primers. SYBR Green PCR system (Roche) was used to carry out real-time PCR with an iQ5 system (BioRad, Hercules, CA, USA). As an internal control, mRNA levels of 18S were measured. The samples were assessed in triplicate. Primers used for qPCR are shown in Table 1.

### 2.8. Western Blot

For protein extraction, cells were lysed on ice for 30 min in TNE buffer (50 mM Tris, 150 mM NaCl, 1 mM EDTA, and 0.5% Triton X100) supplemented with wide-range protease inhibitors (Roche) and lactacystin (Sigma-Aldrich), a specific proteasome inhibitor. Lysates were centrifuged at 14,000× *g* for 5 min. Total protein concentration was determined using Bradford reagent (Bio-Rad) and using BSA as standard, then absorbance was measured at 595 nm using a Glomax Multidetection System (Promega, Madison, WI, USA). Similar amounts of protein from cleared cell lysates were boiled in SDS sample buffer and analyzed by a gradient 4–20% SDS-PAGE under non-reducing conditions (BioRad). Proteins from gels were electro-transferred to nitrocellulose membranes (Amersham, Little Chalfont, UK) followed by immunodetection with primary antibodies such as Anti-Endoglin, P4A4, Anti-VHL, and Anti-VWF from Abcam; Anti-phosphorylated AKT, Anti-AKT, Anti-phosphorylated ERK1/2, and Anti-ERK1/2 from Cell Signaling (Danvers, MA, USA); and Anti-β-actin from Sigma-Aldrich. Following primary antibody incubation overnight at 4 °C, samples were washed and incubated with the corresponding horseradish peroxidase-conjugated secondary antibodies from Dako (Glostrup, Denmark) at RT for 1 h. All antibodies were used at the dilution recommended by the manufacturer. Membranes were developed by chemiluminescence using SuperSignal West Pico Chemiluminescent Substrate (Thermo Fisher Scientific). Bands were measured using the densitometry tool within GIMP 2.10 software (UC Berkeley).

### 2.9. Reactive Oxygen Species Assay

To measure reactive oxygen species (ROS) production, 5 × 10^3^ BOECs (control and VHL) per well were seeded in a 96-well plate. After 72 h, cells were washed with PBS twice and stained by adding 100 μL of diluted DCF-DA solution per well. After 30 min of incubation at 37 °C in the dark, DCF-DA was removed and cells washed. Then, 100 μL/well PBS was added and fluorescence measured (exc: 485 nm/em: 535 nm) (ROS0300, OzBiosciences, San Diego, CA, USA).

### 2.10. Statistical Analysis

Results are presented as mean ± SD. Statistical analyses were performed using the Student’s *t*-test. Statistical significance was defined when *p* < 0.05 (* *p* < 0.05; ** *p* < 0.01, *** *p* < 0.001, **** *p* < 0.0001). Graphs were generated using Graph Pad Prism 8.0 (San Diego, CA, USA).

## 3. Results

### 3.1. Functional Characterization of BOECs Derived from VHL Patients

BOECs were derived from peripheral blood of 5 VHL patients (named V#1–5). Table 2 summarizes the genotype/phenotype correlation and age of the VHL patients from which ECs were derived. V#5 presents an additional rare disease: the patient is heterozygous for CLN5 protein in the neuronal ceroid lipofuscinosis rare disease. Our group hypothesized that the *CLN5* mutation in this VHL patient offers a protective effect, preventing tumor development in those tissues potentially suffering a *VHL* second hit mutation [24].

In addition to these samples, we established two more cultures of BOECs derived from non-VHL control volunteers (healthy donors). BOECs were cultivated following a 5-week isolation and expansion method (Figure 1A), resulting in pure primary ECs cultures from each subject. BOECs were characterized as ECs by flow cytometry using Endoglin (CD105) (endothelial TGFβ co-receptor), VWF, vascular endothelial cadherin (CDH5/VE-Cadherin), and VEGFR2/KDR as endothelial markers. All our BOECs turned positive for all four EC markers (Figure 1B) and negative for CD14 (a macrophage marker, data not shown). Immunofluorescence microscopy further confirmed the EC nature of BOECs by positive staining for VWF, within the Weibel Palade bodies, which is endothelial specific (Figure 1C).

### 3.2. Transcriptome Analysis Derived from RNA Sequencing

Total RNA of the different primary cultures of BOECs, five of VHL patients and two from healthy donors, was isolated and subjected to RNA sequencing. RNA sample integrity and quality of the different samples prior to analysis is shown in Appendix A. The data obtained by RNAseq were subjected to a principal component analysis (PCA), which showed significant differential clustering between the VHL and control groups (Figure 2A). RNAseq approach detected the expression of 13,817 genes, out of the 27,000 genes in the human genome. Differential expression analysis showed statistically significant differences among groups in 1320 genes, with 43 of them having adjusted *p* value < 0.05 (Figure 2B). Among the differentially expressed genes, we found genes involved in membrane organization and adhesion like *CAVEOLIN 1* genes affecting inflammatory response, such as *CCL20*; *IL1β*; and, related to ROS clearing, like superoxide dismutase 2 (*SOD2*).

The differentially expressed genes were participating in pathways relevant to the EC function, which were deregulated in the VHL cells, as shown by gene set enrichment analysis (GSEA) (Figure 2C). The response to oxygen levels is of special interest in VHL, and it is upregulated in the patient group, probably due to HIF-dependent partial activation. Moreover, interestingly endoplasmic reticulum stress response is also upregulated indicating an excess of ROS in these cells (Figure 2C, top). On the other hand, it is remarkable that, among the downregulated genes, we found those involved in the immune response: inflammation, defense-response to other organisms, interferon-γ, and nuclear factor-κB (NF-κB)-triggered pathways (Figure 2C, bottom).

### 3.3. Differential Expression Validation of Genes Relevant to EC Function

Differentially expressed genes relevant to EC function and VHL disease were chosen for validation of the transcriptome analysis. In Figure 3, we show the analysis of Endoglin (*ENG*), endothelial nitric oxide synthase (*eNOS*), and *VHL. ENG*, a TGF-β pathway co-receptor with proangiogenic activity, is significantly downregulated in the RNAseq analysis. On the other hand, *eNOS*, an endothelial-specific gene, crucial for vessel homeostasis, is significantly upregulated in samples from VHL patients compared to control cells. Both results were confirmed by validation of the data by quantitative PCR and western blot. *VHL* shows no differences in the samples from VHL patients as seen by RNAseq, whereas a slight downregulation is detected with RT-qPCR, and a tendency to compensation is observed at protein level (Figure 3A–C). The results could be explained by the fact that four out of the five samples were missense mutations (Table 2), and, therefore, the RNA synthesis does not seem to be affected; a compensatory effect from cell machinery may be present.

Differential gene expression patterns between control and VHL groups were successfully replicated by qPCR in *IL6*, *Il1β*, *CCL20*, and *TNFAIP6* (Figure 4A,B). All these genes are involved in the primary immune response, and their downregulation in VHL ECs is relevant for its functional meaning.

### 3.4. Tubulogenesis and Angiogenesis Assays of VHL versus Control BOECs

Upon isolation and expansion of BOECs, it was visually clear that BOECs derived from VHL patients appeared morphologically different from healthy controls (Figure 5A). While controls appear as elongated and in closer contact, VHL BOECs are less elongated, rather rounded, and sparsely distributed. Angiogenesis can be measured in vitro by two functional assays for ECs: tubulogenesis and wound healing. VHL patient-derived BOECs showed differences compared to the control group. In the tubulogenesis assay, VHL cells displayed a significantly reduced number of closed tubules (Figure 5B). On the other hand, they were slower on average when migrating to cover the discontinuity inflicted in the confluent monolayers or wound (Figure 5C).

### 3.5. VEGF Signaling Pathway

The VEGF signaling pathway appeared upregulated in VHL BOECs, according to the GSEA analysis (Figure 6A). Western blotting was used to check the phosphorylation of regulatory kinases belonging to the VEGFA-VEGFR2 pathway (AKT and ERK1/2) (Figure 6B).

### 3.6. VHL ECs Have Higher Expression of ROS

Figure 7A shows a table collecting genes involved in the ROS metabolism, which are downregulated in VHL ECs. Accordingly, a measure of ROS was made from in vitro cultures of control and VHL-derived ECs. In Figure 7B, a representative histogram of the results is depicted. As expected, and supporting the RNAseq results, we found that VHL ECs have on average almost double the amount of ROS than their non-VHL counterpart.

### 3.7. VHL Gene Expression in VHL Patients’ ECs

No differences in *VHL* gene expression were found between the patients and the control group, despite the heterozygosis of the patients in the *VHL* gene. This result could be explained by the type of mutation these patients present: four out of five present missense mutations (Table 1, Figure 3). Only in sample 4, where the mutation affects the splicing site of the DNA, the RNA levels show a lower tendency, although not significant (Appendix A). In order to visualize the translational effect of the patients’ mutations, we mapped them on the 3D structure of pVHL, highlighting the changes in amino acids (Figure 8A,B). The altered amino acids fall within areas of close contact between the different subunits, which could affect its proteolytic activity and the assembly of a functional proteasomal complex. Proper folding of the protein, an altered interaction with the substrates, or assembly of the E3 ligase complex, which most likely lead to its loss of function. Further computational analysis should be performed to elucidate the definite cause of how those mutations impair protein function.

A misfolding of the protein could explain the differential distribution of pVHL in sample V1 (p.R167Q). Confocal staining of ECs shows that pVHL accumulates in the perinuclear region. This result suggests a likely aggregation of pVHL in endoplasmic reticulum cysts. Conversely, control ECs show a more heterogeneous distribution of pVHL throughout the cytoplasm (Figure 8C). We believe that these differences in cellular pVHL distribution are caused by the mutations in pVHL present in this patient.

## 4. Discussion

The main tool used to develop the work here presented is human-derived EC primary cultures, a novel approach in VHL patients. The isolation and growth of BOECs constitutes in itself a major limitation due to (1) the scarcity of endothelial precursor cells (EPCs) in peripheral blood; and (2) the difficulty and time-consumption of expanding these cells, which grow at a lower rate than other primary cultures. Despite these limitations, BOECs constitute the closest cell model to approach the cell physiology of VHL patients outside the tumoral foci and are, therefore, of great research value.

Human periphery blood contains, on average, 1–3 EPCs per mL in basal conditions [25,26]. These are available for recruitment in case of endothelium damage or hypoxic stress, when VEFG signaling pathway promotes angiogenesis [27]. The upregulation of the VEGF pathway in VHL-derived BOECs (Figure 6) may be the origin of a higher success rate achieved when growing BOECs from VHL peripheral blood patients (data not shown); higher levels of endogenous VEGF stimulate the release of EPCs from the bone marrow [28], thus facilitating ECs’ culture growth. The following major EC functions were found altered in VHL-derived BOECs when RNAseq was performed in five VHL different primary cultures versus control cells: cell adhesion, angiogenesis, migration, immune response, and metabolism (ROS).

### 4.1. Cell Adhesion

We have observed that VHL-derived BOECs grow slower than those derived from healthy subjects, enduring fewer subculture passages and with aging appearance after a few passages, 3–4 on average. BOECs from VHL patients do not seem to establish tight contacts with each other, appearing morphologically different and less elongated than control cells (Figure 5A). This could be explained by a downregulation of cell adhesion molecules (CAMs), key for ECs contact, like intercellular adhesion molecule 1 (ICAM1), vascular cell adhesion molecule 1 (VCAM1), Fibronectin (FN1), and CD44. Higher levels of Endomucin (EMCN) (Appendix A) would also be unfavorable to cell–cell interaction as this sialoglycoprotein has been shown to interfere with focal adhesion complex assembly, inhibiting cell–extracellular matrix interaction [29]. Altogether, this pattern of expression is detrimental to cell adhesion, a fundamental condition for the homeostasis of vascular biology [30].

### 4.2. Angiogenesis and Cell Migration

The dysregulation of key adhesion genes not only correlates with an aberrant morphology but also with a proper angiogenesis visualized in vitro by the tubulogenesis assay (Figure 5B) and delay in wound healing performance (Figure 5C). Cell adhesion and angiogenesis are closely interrelated, partly due to the role of CAMs like VCAM1 in angiogenesis [31] and shared signaling pathways such as VEGF A-VEGFR2 [27]. As a result of a VEGF-A/VEGFR-2 upregulated pathway (Figure 6A), VHL-derived BOECs show higher phosphorylation levels in ERK1/2 (Figure 6B). On the other hand, cell–extracellular matrix interactions via Endoglin-mediated signaling have been shown crucial for angiogenesis [32,33]. For this reason, the decreased levels of *ENG* observed by both RNAseq and qPCR in VHL-derived BOECs (Figure 4) correlate with the functional defects in tubulogenesis and wound healing assays observed (Figure 5B,C).

### 4.3. Immune Response

To the best of our knowledge, there is no record of VHL patients being immunocompromised; however, our RNAseq results suggest otherwise. This might be worth validating in vitro. The NF-κB has been extensively studied, showing special involvement of the immune system because of its role in regulating both the innate and adaptive immune responses [34]. *NF-κB* is activated by a wide range of stimuli, including pro-inflammatory cytokines like TNF-α. Downregulation of *TNF-α* and *ENG* genes (Appendix A and Figure 3) correlates with a downregulation of the NF-κB pathway (Figure 2C) and its gene targets (Figure 4) in VHL-derived BOECs. Important targets of NF-κB in immune response, like matrix metalloproteinases (MMPs), cylo-oxygenase 2 (COX2), and nitric oxide synthase (NOS3) [35], are all downregulated in VHL-derived BOECs (Appendix A).

### 4.4. Cell Metabolism and ROS

The results obtained in regard to cell respiration correlate with the pVHL haploinsufficiency that VHL patients suffer in their cells, due to their inherited mutation. Excess HIF levels are evidenced by an upregulation of HIF targets like *eNOS3* [36] (Figure 5) and *CAVEOLIN 1* (CAV1) [37] (Appendix A). VHL patients’ endothelium display a partially constitutively activated hypoxia program despite normoxic levels, which in itself causes ROS accumulation [38]. Additionally, VHL patients show downregulation of key antioxidant enzymes in charge of ROS homeostasis [39], seen in our RNAseq analysis, with lower transcript levels of *SOD2* and glutathione reductase (*GSR*) as the most remarkable cases (Appendix A). The increase of ROS (Figure 7) due to the state of pseudo-hypoxia, coupled with under-expression of antioxidant enzymes, may predispose the endothelium of VHL patients to ageing conditions, derived from prolonged oxidative stress. Hypoxia-induced mitochondrial autophagy is the cells’ attempt to reduce mitochondrial ROS production under a prolonged hypoxia stress and has been shown to be HIF-1 dependent [40]. The detected upregulation in the autophagy pathway (Figure 2C, Appendix A), as seen by higher transcript levels of autophagy related protein 9B (*ATG9B*) and death-associated protein kinase 1 (*DAPK1*) (Appendix A), is another consequence of a sustained hypoxia state that dysregulates cell metabolism in VHL-derived BOECs.

HIF-1 has also been described to activate the transcription of genes coding for glycolytic enzymes, shifting the energy cell balance towards glycolysis [40]. Upregulation of hexokinase 2 (*HK2*) and pyruvate kinase (*PKM*) in VHL-derived BOECs (Appendix A) is indicative of the cell metabolism shift in these patients, caused by the prolonged state of pseudo-hypoxia due to the constitutive partial activity of HIF-1.

In summary, the excess of HIF in cells from VHL patients, caused by their pVHL haploinsufficiency, is evidenced by an upregulation of pathways involving key HIF targets. The ECs derived from VHL patients undergo a dysregulation of genes involved in cell adhesion, angiogenesis, migration, immune response, and metabolism. The altered transcriptome profile of VHL-derived BOECs correlates with the aberrant morphology and deficient cell function detected in vitro. These results illustrate a partially dysfunctional endothelium in VHL patients that is likely more fragile, in the event of a second hit mutation, facilitating tumor development. All these states are illustrated by the model depicted in Figure 9.

## Figures and Tables

**Figure 1 cells-10-02313-f001:**
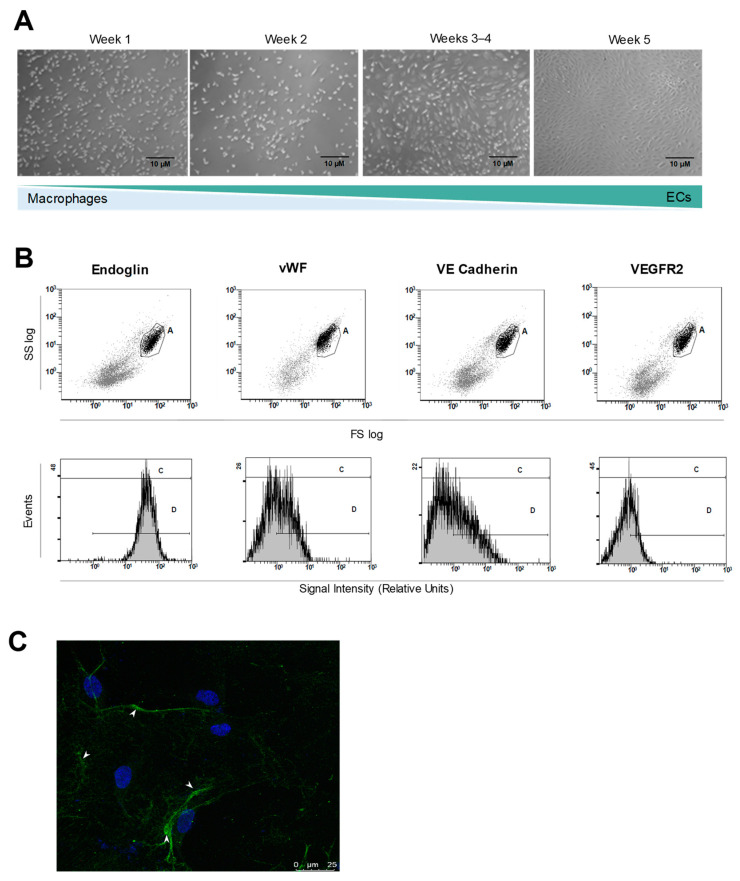
(**A**) Isolation of ECs derived from periphery blood samples (BOECs). Photograph sequence of BOECs culture growth and expansion, weeks 1 through 5. As days pass and cell medium is changed, macrophages die and leave room for ECs expansion. These images are representative of EC culture isolation from both patients and healthy donors. (**B**) Flow cytometry characterization of the ECs obtained after week 5 of expansion, using EC markers: Endoglin, VWF, VE Cadherin, and VEGFR2. Graph shows expression index of all four markers, as product of the intensity and the % of positive cells. (**C**) VWF localization inside the isolated ECs, in Weibel Palade bodies, as seen by confocal microscopy (green). The nuclei are in blue from DAPI staining.

**Figure 2 cells-10-02313-f002:**
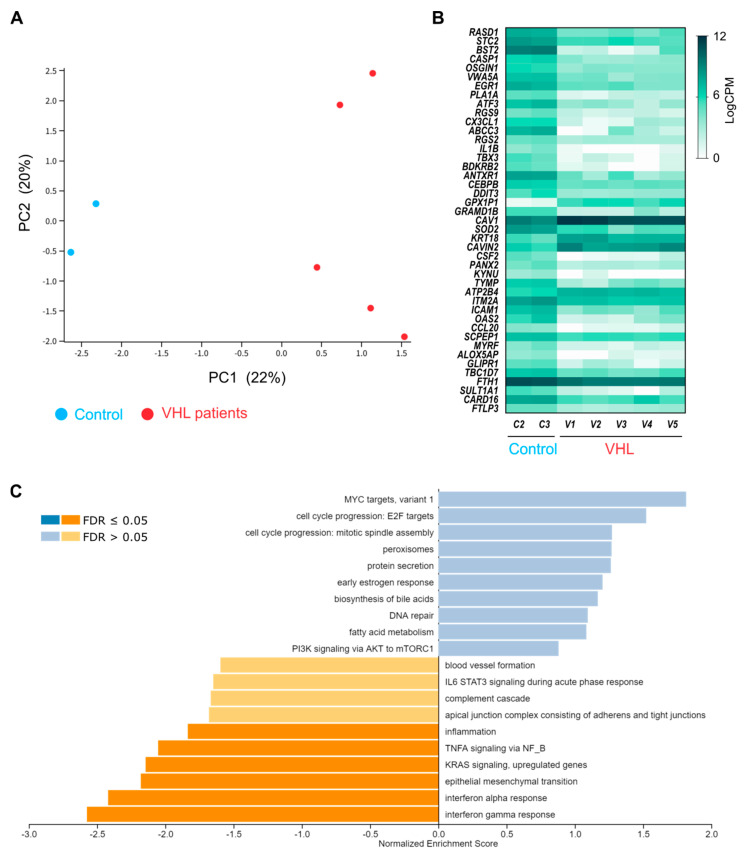
(**A**) Principal component analysis (PCA) showing the drift in transcriptome expression between control subjects (blue) and VHL patients (red). (**B**) Heat map of the logCPMs from control and VHL groups, taking only genes with adjusted *p* value < 0.05 in the PCA analysis. (**C**) Gene set enrichment analysis (GSEA) by Hallmarks of Cancer geneset. Blue shows over-represented genes (categorized by biological function) in VHL patients, as compared to control subjects. Orange shows unrepresented gene-sets in VHL patients.

**Figure 3 cells-10-02313-f003:**
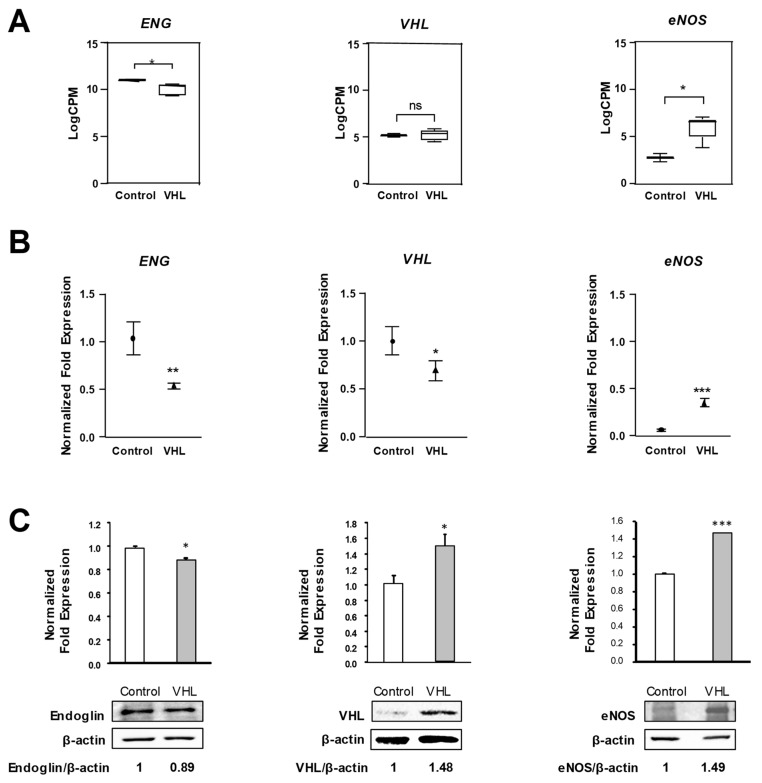
RNAseq validation for *Endoglin*, *VHL*, and *eNOS*. (**A**) RNAseq results derived of primary cultures of ECs from control (*n* = 2) and VHL patients (*n* = 5). (**B**) qPCR results of *Endoglin*, *VHL*, and *eNOS* from different controls (*n* = 2) and VHL ECs (*n* = 3). (**C**) Western-blot analysis of Endoglin, VHL, and eNOS protein levels from primary ECs, normalized to β-actin. Densitometric quantification of Endoglin/β-actin, VHL/β-actin, and eNOS/β-actin ratios are shown. Results shown are representative of three different experiments. Error bars denote ± SEM. Student’s *t*-test: * *p* < 0.05; ** *p* < 0.01; and *** *p* < 0.001.

**Figure 4 cells-10-02313-f004:**
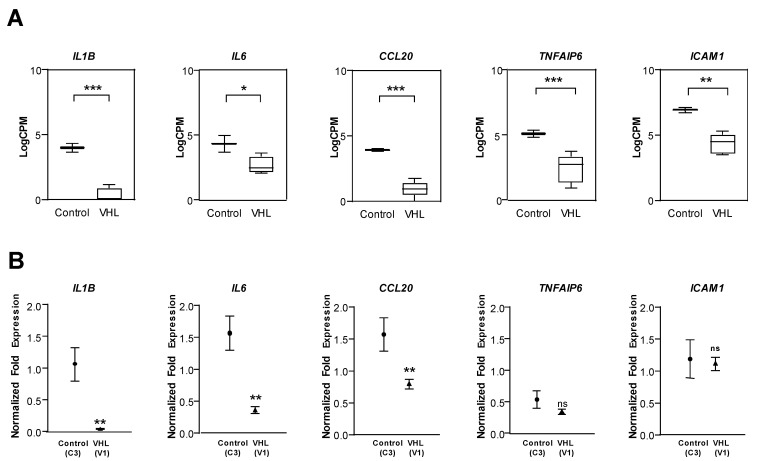
RNAseq validation for different genes involved in the inflammatory response: *IL1B*, *IL6*, *CCL20*, *TNFAIP6*, and *ICAM1*. (**A**) RNAseq results of primary cultures derive from ECs from control (*n* = 2) and VHL patients (*n* = 5). (**B**) qPCR results derived from RNA samples obtained from controls (*n* = 2) and VHL primary cultures (*n* = 3) of ECs. The results shown are representative of three independent experiments. Error bars denote ± SEM. Student’s *t*-test: * *p* < 0.05; ** *p* < 0.01; and *** *p* < 0.001.

**Figure 5 cells-10-02313-f005:**
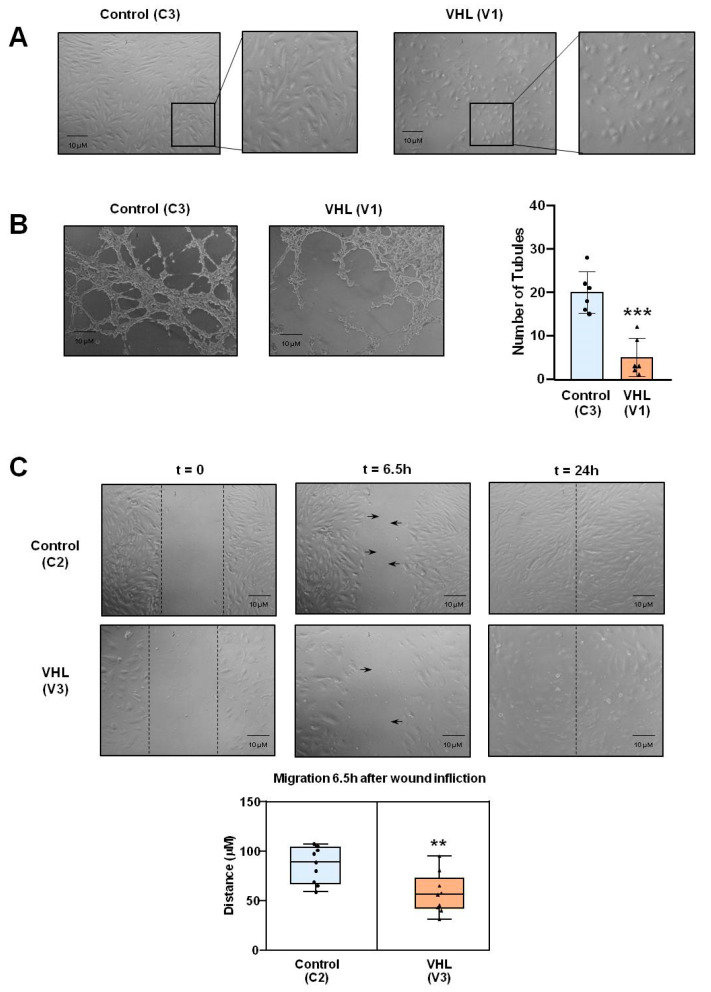
Morphological and functional differences between ECs derived from either healthy subjects or VHL patients. (**A**) Control and VHL ECs photographed under the microscope once confluence levels were reached. (**B**) Tubulogenesis assay performed on ECs from control vs. VHL groups, photographed 3 h after cell seeding on Matrigel-matrix-coated wells. (**C**) Wound healing assay performed on ECs from control vs. VHL groups, photographed 6.5 h after wound infliction on confluent wells. Error bars denote ± SEM. Student’s *t*-test: ** *p* < 0.01; and *** *p* < 0.001.

**Figure 6 cells-10-02313-f006:**
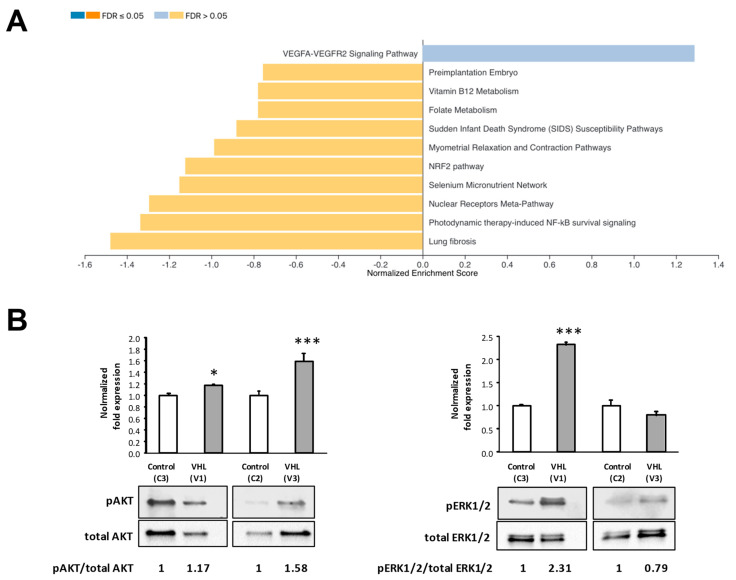
(**A**) VEGF signaling pathway appeared upregulated, according to the GSEA analysis. (**B**) ERK1/2 and AKT phosphorylation assessment by Western Blot. Densitometric quantification of phosphorylated/total protein ratios are shown for control subject (*n* = 2) or VHL patients (*n* = 2). Error bars denote ± SEM. Student’s *t*-test: * *p* < 0.05 and *** *p* < 0.001.

**Figure 7 cells-10-02313-f007:**
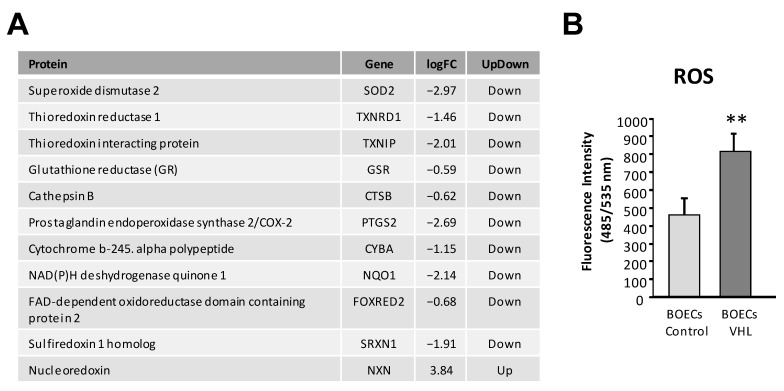
(**A**) ROS metabolism modulation in RNAseq analysis in VHL ECs. (**B**) ROS levels in control and VHL cells, showing a significant increase in VHL samples. Error bars denote ± SEM. Student’s *t*-test: ** *p* < 0.01.

**Figure 8 cells-10-02313-f008:**
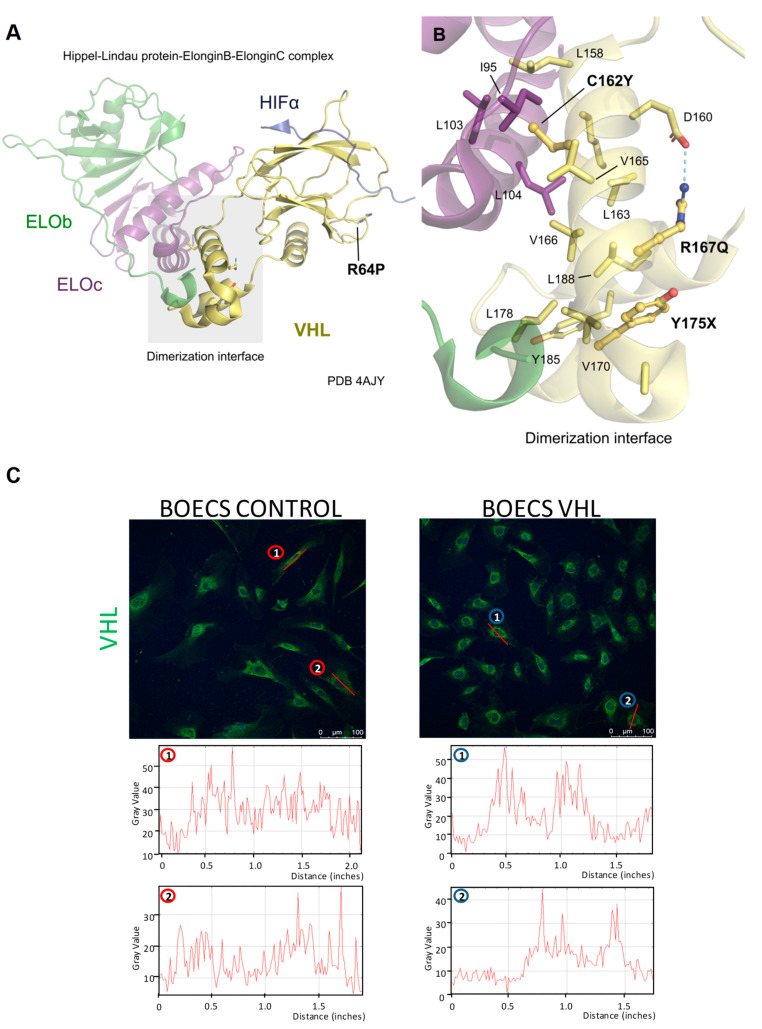
(**A**) 3D complex structure (PDB ID: 4AJY) of the von Hippel–Lindau protein (yellow), Elongin B (green), and Elongin C (purple), bound to Hif1- alpha peptide (depicted in dark blue), highlighting the heterodimerization interface of this complex in a gray box and the position of the R64P mutation. (**B**) Inset of the VHL-ELOb/c interface with their respective involved amino acids are represented as sticks, while the relevant mutation positions are represented with spheres and labelled in bold. (**C**) Confocal staining of sample V1 (p.R167Q) shows the distribution of pVHL in the cell cytoplasm, distributed in the perinuclear region in VHL cells and throughout the cell in control ones.

**Figure 9 cells-10-02313-f009:**
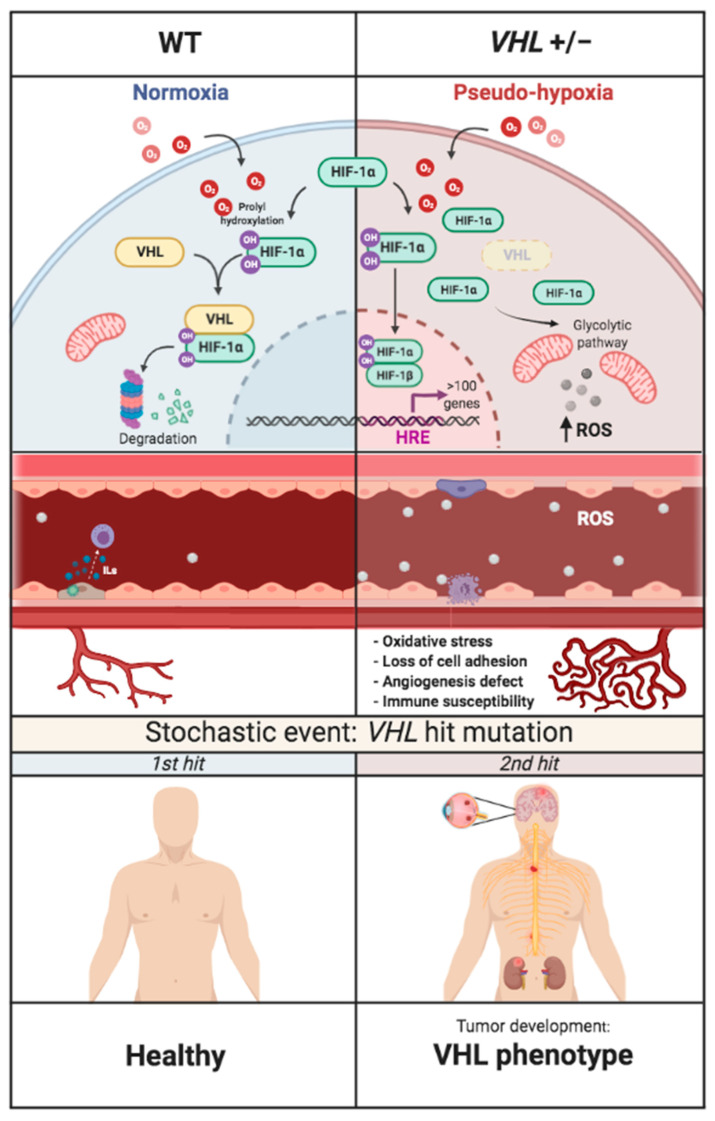
Flow chart of the development of VHL disease. The absence of pVHL promotes the accumulation of HIF-1 in the cell and thus a state of permanent pseudo-hypoxia, manifested by loss of adhesion capacity, defective angiogenesis, immune susceptibility, and altered metabolism, inducing prolonged oxidative stress. The endothelium of these patients is partially dysfunctional and more susceptible to tumor development. All this precedes the second hit mutation in *VHL* that triggers tumorigenesis.

**Table 1 cells-10-02313-t001:** Primers used for q-PCR assays.

Gene	Fwd 5′–3′	Rev 5′–3′
18S	CTCAACACGGGAAACCTCAC	CGCTCCACCAACTAAGAACG
ENG	GCCCCGAGAGGTGCTTCT	TGCAGGAAGACACTGCTGT
VHL	ATCCGTAGCGGTTGGTGA	CTCACGGATGCCTCAGTCTT
NOS3	GACCCTCACCGCTACAACAT	CCGGGTATCCAGGTCCAT
CCL20	GCTGCTTTGATGTCAGTGCT	CAGTCAAAGTTGCTTGCTGCT
IL1β	CTGTCCTGCGTGTTGAAAGA	TTGGGTAATTTTTGGGATCTACA
IL6	CAGGAGCCCAGCTATGAACT	GAAGGCAGCAGGCAACAC
ICAM1	CCTTCCTCACCGTGTACTGG	AGCGTAGGGTAAGGTTCCTTGC
TNFAIP6	GGCCATCTCGCAACTTACA	GCAGCACAGACATGAAATCC

**Table 2 cells-10-02313-t002:** Genotypes and phenotypes of the VHL samples.

Patient	Location	c.DNA	Protein Change	Clinical Symptoms
V#1	exon 3	c.500G > A	p.R167Q	CNS-HB, renal carcinoma
V#2	exon 3	c.485G > A	p.C162Y	CNS-HB
V#3	exon 3	c.525C > A	p.Y175X	Pancreatic cysts and renal carcinoma
V#4	Sp intron 2	c.463, +2A > C		Retinal-HB, cerebellum, spinal cord, and renal carcinoma
V#5	exon 1	c.191G > C	p.R64P	No tumors

## Data Availability

Reported results can be found in the files of Centro de Investigaciones Biológicas Margarita Salas (CIB, CSIC). RNAseq data is available as GEO dataset GSE181417.

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
