# Peer review of "The Endothelial Landscape and Its Role in Von Hippel–Lindau Disease"

_cells, 2021, doi:10.3390/cells10092313_

Round 1
Reviewer 1 Report
This work is interesting and well structured. However there are some major concerns that the authors should clarify/address.
- I do not understand why only two controls (healthy donors) were recruited. Most of the control experiments have been done using data from only those two control patients and this cannot be considered statistically significant. I understand the difficulty in recruiting patients suffering from a rare disease, but that does not justify the recruitment of only two controls, which are healthy donors that could be easily found. Also, in figure 3 and 4 the legend states that the results shown are representative of three independent experiments. Does that mean that different sets of experiments have been done in cells from the same control donor (as you have only 2 of those)? In those two figures it would be good to see all data point in the graphs.
- Figure 1C: at what week of growth was the picture taken? Why there are so few cells and why they are not all positive for green?
- Figure 2 is very poor quality, it looks like it has been copied and paste from a pdf. Figures need to be prepare with an appropriate software where the writing is clear and focused.
- In the western blots we should see a proper quantification, not just a ratio number. There should be a histogram with the different points (at least 3 per condition), the mean and the SD or SE. If those data are only from one control and one patients, they cannot be considered conclusive.
- In the methods a statistical section is missing. We need to know how data have been analysed and which statistic tests have been used.
- In all the figure legends statistical significance must be addressed, i.e. clarify what test has been used and add the p values as for instance *p<0.05 **p<0.001 etc.
Author Response
Referee 1
This work is interesting and well structured. However there are some major concerns that the authors should clarify/address.
Thank you very much for your appreciation of the work. We have tried to address all your comments to improve the manuscript
I do not understand why only two controls (healthy donors) were recruited. Most of the control experiments have been done using data from only those two control patients and this cannot be considered statistically significant. I understand the difficulty in recruiting patients suffering from a rare disease, but that does not justify the recruitment of only two controls, which are healthy donors that could be easily found.
Thank you for your point; we should have explained this in more detail.
The experimental design was conceived so that every time we recruited a patient, we got in parallel a control. We tried to grow VHL and control cells at the same time. This was done 8 times. While the percentage of success was 5/8 in VHL, we only got 2/8 control cells growing. It may be surprising that we were more successful getting the cultures from VHL patients than from Controls. However, this seems to have an explanation. These endothelial cultures come from Circulating Endothelial Precursor Cells. The mobilization of these cells from the bone marrow is higher in VHL patients than in controls, probably because VHL patients have high levels of VEGF due to active angiogenesis in the points of tumoral growth [Albiñana et al, 2017, and Cuesta et al, 2019].
Our purpose was to get the endothelial cell cultures at passage 2 for the RNAseq processing at the same time. Therefore, once we got 5 VHL and 2 Controls for the analysis, we thought it could be enough if the results were sufficiently homogenous since everything was done in parallel. You are right, having more replicas increases the predictive power of the transcriptomes, however given the limitation (collection), we chose to proceed with the replica number we had and spend more time in the validation of the genes by qPCR and blotting. On the other hand, we were very strict with the transcriptome analysis data, and fortunately as shown in Figure1, the two controls and the VHL samples are quite concordant.
The work from Schurch et al., (2016, RNA. 2016 Jun;22(6):839-51. doi: 10.1261/rna.053959.115) indeed shares the reviewers concerns regarding the number of replicas. They demonstrated that the lower numbers of replicas would imply a high false positive rate, as well as hamper the prediction of lowly modulated DEGs. However, they also show that with some degree of confidence the use of EdgeR tool combined with careful post-processing analyses can provide an adequate framework for these challenging cases.
In addition to this explanation, we must emphasize on the difficulty in getting this type of cultures. This is not a technique currently used. Getting primary endothelial cells cultures from 50 ml of peripheral blood requires expertise and patience. In the literature, there are not many papers using this type of cultures. In fact, the technique was published for the first time by Dr. Hebbel’s lab (Li et al, 1999) starting from 100 ml of peripheral blood from control volunteers, and we visited his lab to learn the technique. We published the technique, adapted with half the volume of blood (50 ml) for Hereditary Hemorrhagic Telangiectasia (HHT) patients in 2005 in a paper co-authored by Dr. Hebbel and Dr. Nguyen [(Fernandez-L A, et al. Blood outgrowth endothelial cells from Hereditary Haemorrhagic Telangiectasia patients reveal abnormalities compatible with vascular lesions. Cardiovasc Res. 2005 Nov 1;68(2):235-48. doi: 10.1016]
This is the first time that ECs from VHL patients have been studied, an important step to understand the disease.
Also, in figure 3 and 4 the legend states that the results shown are representative of three independent experiments. Does that mean that different sets of experiments have been done in cells from the same control donor (as you have only 2 of those)?
Thank you for your precision. In fact, the results shown include three different experiments, where we have used the two different controls we had, and different VHL patients.
In those two figures it would be good to see all data point in the graphs.
The point is well taken. The graphs shown are box-whiskers plots visually displaying the data distribution of all the points. The lines extending from the boxes (“whiskers”) indicate variability outside the upper and lower quartiles.
Figure 1C: at what week of growth was the picture taken? Why there are so few cells and why they are not all positive for green?
To get a confluent monolayer of around 100.000 endothelial cells as passage 0 directly from blood, it takes around 1 month (from the day of blood extraction). Since we started with the RNAseq at passage 2 (around 5x106 cells), the cells took in total 6 weeks after blood extraction to be ready for the RNAseq.
The characterized cells come from passage 3, around 1 week of culture after thawing passage 2 stored cells, and plating them.
The picture is a confocal microscopy section, thus some of the cells seem to have brighter staining than others, but all are positive as supported by flow cytometry. The plot (Fig 1B second plot) for von Willebrand Factor (vWF) stained and analyzed by flow cytometry shows that ECs are positive, although with different intensity degree as reflected by the distribution.
Figure 2 is very poor quality, it looks like it has been copied and paste from a pdf. Figures need to be prepare with an appropriate software where the writing is clear and focused.
We apologize for the poor quality, we have now fixed this problem.
In the western blots we should see a proper quantification, not just a ratio number. There should be a histogram with the different points (at least 3 per condition), the mean and the SD or SE. If those data are only from one control and one patients, they cannot be considered conclusive.
Thank you very much for your precision. We have included the bar histograms with the SD. These data are representative from several patients and 2 different controls.
In the methods a statistical section is missing. We need to know how data have been analysed and which statistic tests have been used.
In the methods, we have included in the 2.10 section the statistical tests performed in qPCRs and WB. For the analysis of the RNAseq, the methodology and programs used are explained in section 2.6
In all the figure legends statistical significance must be addressed, i.e. clarify what test has been used and add the p values as for instance *p<0.05 **p<0.001 etc.
Thank you very much. P values of the statistical test were missing in Figures 3, 4, 5 and 6 . We have added them now.

Reviewer 2 Report
The authors demonstrated the characterization and transcriptome analysis of VHL-derived blood outgrowth endothelial cells (BOECs). They performed the RNA sequencing of BOECs from 5 VHL patients and 2 healthy donors, and analyzed and validated differential expressed genes relevant to EC function. They also performed tubulogenesis and angiogenesis assay of BOECs from patients and healthy donors. They found upregulated VEGF signaling and increased ROS in VHL-patients.
Comments:
Figure 1A, the authors showed the EC culture but EC marker and macrophage marker staining are missing.
Figure 1, based on the text, these images are representative images for all 5 patients and 2 healthy donors? Please clarify in the figure legend. Also, is there any difference of EC numbers between VHL-samples and control sample?
Figure 3A, 3B, 4A, 4B, please add the n numbers for each group in the figure legend.
Figure 3C, 6B: quantification graphs with error bars and p values for each protein: Endoglin, VHL and eNOS in 3C, pAKT, pERK1/2 in 6B are missing.
Author Response
Referee 2
The authors demonstrated the characterization and transcriptome analysis of VHL-derived blood outgrowth endothelial cells (BOECs). They performed the RNA sequencing of BOECs from 5 VHL patients and 2 healthy donors, and analyzed and validated differential expressed genes relevant to EC function. They also performed tubulogenesis and angiogenesis assay of BOECs from patients and healthy donors. They found upregulated VEGF signaling and increased ROS in VHL-patients.
Comments:
Figure 1A, the authors showed the EC culture but EC marker and macrophage marker staining are missing.
Thank you for your comment. The characterization of cells with EC markers is shown in Fig 1, panels B and C. In B 4 different endothelial markers are shown: Endoglin, von Willebrand Factor, VE-cadherin and the receptor 2 of VEGF (VEGF-R2). In addition, as von Willebrand factor is intracellular, a confocal microscopy picture is shown in panel 1C to visualize the staining.
A marker for macrophages CD14 was used in flow cytometry, but the result of was negative (data not shown).
Figure 1, based on the text, these images are representative images for all 5 patients and 2 healthy donors? Please clarify in the figure legend. Also, is there any difference of EC numbers between VHL-samples and control sample?
Thank you for your comment, we have added to the legend of Figure 1 that the images are representative for all the 5 patients and 2 healthy donors.
For the purposes of characterization, we tried to use the same number of VHL and controls since we were counting equivalent numbers. There are no differences in cell number.
Figure 3A, 3B, 4A, 4B, please add the n numbers for each group in the figure legend.
Thank you for your comment. At the end of each Figure 3 and 4 we mention that “Results shown are representative of 3 different experiments”. Nonetheless, we have now added the N in each legend for further clarification.
Figure 3C, 6B: quantification graphs with error bars and p values for each protein: Endoglin, VHL and eNOS in 3C, pAKT, pERK1/2 in 6B are missing.
Thank you very much, now we have included error bars and p values for each protein in western blots (Figure 3 and 6).

Round 2
Reviewer 1 Report
Thank you for addressing all my comments. I do understand the challenge of growing EPCs as I did it for many years and it heavily relies on the specific donor. For the future it might be good to choose female volunteers, as they have more EPCs going around that are needed to repair the uterus after the menstrual cycle.
Reviewer 2 Report
Questions are addressed.